# Tuning the Envelope Structure of Enzyme Nanoreactors for In Vivo Detoxification of Organophosphates

**DOI:** 10.3390/ijms242115756

**Published:** 2023-10-30

**Authors:** Tatiana Pashirova, Zukhra Shaihutdinova, Dmitry Tatarinov, Milana Mansurova, Renata Kazakova, Andrei Bogdanov, Eric Chabrière, Pauline Jacquet, David Daudé, Almaz A. Akhunzianov, Regina R. Miftakhova, Patrick Masson

**Affiliations:** 1Arbuzov Institute of Organic and Physical Chemistry, FRC Kazan Scientific Center, Russian Academy of Sciences, Arbuzov Str., 8, 420088 Kazan, Russia; shajhutdinova.z@mail.ru (Z.S.); datint@iopc.ru (D.T.); abogdanov@inbox.ru (A.B.); 2Institute of Fundamental Medicine and Biology, Kazan Federal University, 420008 Kazan, Russia; chirkova.milana@yandex.ru (M.M.); kazakova.renata@mail.ru (R.K.); aaahunzyanov@kpfu.ru (A.A.A.); regina.miftakhova@gmail.com (R.R.M.); 3Gene&GreenTK, 19–21 Boulevard Jean Moulin, 13005 Marseille, France; eric.chabriere@univ-amu.fr (E.C.); pauline.jacquet@gene-greentk.com (P.J.); david.daude@gene-greentk.com (D.D.); 4IRD, APHM, MEPHI, IHU-Méditerranée Infection, Aix Marseille Université, 19–21 Boulevard Jean Moulin, 13005 Marseille, France

**Keywords:** enzyme nanoreactor, polymersomes, organophosphate poisoning, phosphotriesterase, prophylaxis, post-exposure treatment, pharmacokinetics, immune response

## Abstract

Encapsulated phosphotriesterase nanoreactors show their efficacy in the prophylaxis and post-exposure treatment of poisoning by paraoxon. A new enzyme nanoreactor (E-nRs) containing an evolved multiple mutant (L72C/Y97F/Y99F/W263V/I280T) of *Saccharolobus solfataricus* phosphotriesterase (PTE) for in vivo detoxification of organophosphorous compounds (OP) was made. A comparison of nanoreactors made of three- and di-block copolymers was carried out. Two types of morphology nanoreactors made of di-block copolymers were prepared and characterized as spherical micelles and polymersomes with sizes of 40 nm and 100 nm, respectively. The polymer concentrations were varied from 0.1 to 0.5% (*w*/*w*) and enzyme concentrations were varied from 2.5 to 12.5 μM. In vivo experiments using E-nRs of diameter 106 nm, polydispersity 0.17, zeta-potential −8.3 mV, and loading capacity 15% showed that the detoxification efficacy against paraoxon was improved: the LD_50_ shift was 23.7xLD_50_ for prophylaxis and 8xLD_50_ for post-exposure treatment without behavioral alteration or functional physiological changes up to one month after injection. The pharmacokinetic profiles of i.v.-injected E-nRs made of three- and di-block copolymers were similar to the profiles of the injected free enzyme, suggesting partial enzyme encapsulation. Indeed, ELISA and Western blot analyses showed that animals developed an immune response against the enzyme. However, animals that received several injections did not develop iatrogenic symptoms.

## 1. Introduction

Medical applications of enzymatic nanoreactors and nanodevices [1,2,3,4,5] are rapidly being developed. Particularly important achievements have been made in the therapy of inflammatory processes and tissue regeneration, where reactive oxygen species can be scavenged in enzyme cascade reactions [6]. Coupled enzyme reactions have also been exploited in nanoreactors for the in vivo detoxification of alcohol [7]. Neutralization of xenobiotics in nanodevices is indeed another active field, expanding the use of bioscavengers as therapeutic tools against strong toxicants [8,9,10,11], for example, alcohol poisoning [12,13], reduction of chemotoxicity during tumor therapy [14], or neutralization of toxic pathogens [15].

Nanodevices for detoxification are aimed at removing toxic molecules from biological tissues. In the presence of therapeutic nanodevices, the toxicant diffuses inside the nanobody, where it is inactivated. Conversely, drug delivery systems focus on encapsulating drugs and their release under the influence of external factors, for example [16,17,18,19]. The creation of nanoreactors for trapping toxicants involves the complete sealing of enzyme molecules inside an nR body, and must ensure long-term circulation in the body to trap and neutralize toxicants present in the bloodstream and may be slowly released from cellular and organ depot sites, e.g., OP molecules may accumulate in fat from where they are subsequently slowly released into the bloodstream again [2,20,21]. Attempts to use OP-reacting enzymes and their encapsulated forms for stoichiometric or catalytic inactivation of OPs have been undertaken for more than 20 years and have led to very effective formulations [22,23] and novel nanoscavenger devices for OP detoxification [24,25]. However, important issues, such as the operational stability of nano-formulations and fate in organisms after administration (degradation, elimination, and immunogenicity) for the safe use of non-human enzymes have not been completely solved [26,27,28]. Furthermore, given that detoxification reactions take place within stable enzyme-containing nanobodies, little attention has been paid to date to the mechanistic aspects of the reaction. Only recently have the formal mechanisms and parameters that control reaction kinetics inside enzyme nanoreactors started to be investigated [29,30].

In a previous work [31], we showed that the injection of E-nRs containing a high concentration of an evolved multiple mutant of the hyperthermophilic archaea *Saccharolobus solfataricus* phosphotriesterase-like lactonase (PLL) [32] can protect mice against multiple LD_50_ of paraoxon (POX) (up to 16.6 LD_50_ as the sole prophylactic mean). This SsoPox variant was shown to efficiently degrade analogs of chemical warfare nerve agents [33]. Other SsoPox variants were previously reported to be active against tabun, sarin, soman, and cyclosarin [34,35]. In addition, post-exposure injection of these E-nRs was also used as the sole treatment against POX poisoning up to 9.6 × LD_50_. Complete enzyme sealing inside a nanoreactor and long-term circulation in the body are necessary for the effective trapping and neutralization of toxicants. Long-term circulation is important, as toxicants can accumulate in fats, where they are slowly released. Moreover, long-term protection against toxicants is needed for effective prophylaxis. Polyethylene glycol (PEG) guarantees stealth behavior for improved circulatory properties of nanoparticles after their systemic administration. PEG-containing copolymers on the surface of nanoparticles are able to repel the absorption of opsonin proteins via steric repulsion forces, thereby blocking and delaying the first step in the opsonization process [36]. However, this bionanotechnology is not simple. Indeed, nanoparticle action and transit in the body depend on multiple factors, including the PEGylation density or the presence of both short and long PEG chains [37]. However, there are not enough data about the influence of physicochemical properties of polymersomes and their surface chemistry on the biodistribution of these nanobodies [38,39]. The biodistribution of polymersomes was studied for PEG-PPS systems, depending on their morphologies discovered so far [40,41,42].

In the present work, physicochemical tuning of nanoreactor morphology was developed and lower concentrations of the PTE mutant were encapsulated. Tail-vein-administered enzyme-loaded nanoreactors were capable of conferring higher prophylaxis and higher therapeutic index to mice sub-cutaneously (s.c.) when challenged by multiple LD_50_ of POX. Labyrinth tests provided evidence for the neurobehavioral safety of this therapeutic approach. The pharmacokinetics was investigated and raised important questions about the existence of a possible enzyme “corona” on the outer surface of nanoreactors. Lastly, the overall immune response to injected enzyme-loaded nanoreactors was analyzed using ELISA and Western blotting.

## 2. Results and Discussion

### 2.1. Synthesis of Amphiphilic Block Copolymers for Construction of Nanoreactors

In this work, we focused on the synthesis of new polysulfide-polyethylene glycol (PEG-PPS) di-block copolymers according to the synthesis presented in Figure 1. The polymer structures were confirmed using physicochemical methods, NMR ^1^H, ^13^C, and IR (Appendix A).

Unlike the three-block copolymers used in our previous work [31], there are no S-S bonds in the structures of the **1a** and **1b** PEG-PPS. Newly synthesized **1a** and **1b** are not sensitive to external stimuli. PEG di-block copolymers have greater durability and can maintain a denser PEG brush, and thus, potentially a stronger stealth effect [43]. In addition, the idea was to explore nanoreactor construction with different polymer morphologies. It was assumed that polymers **1a** and **1b** have different morphologies. According to the literature, the hydrophilic fraction (f_PEG_) is a benchmark for determining the morphologies of self-assembling PEG-PPS structures (spherical micelle, worm-like micelle and polymersomes) [44]. Thus, the f_PEG_ was 0.44 for **1a**, suggesting the formation of spherical micelles. **1b** with f_PEG_ = 0.27 indicates the formation of polymersomes [45,46]. The f_PEG_ values of **1a** and **1b** were found using Mw(PEG)/Mw(PEG) + Mw(PPS) [47]. The calculation of the PEG/PPS ratio was done by comparing the integral intensity of the PPS methyl group protons with that of the methoxy group protons of mPEG from the ^1^H NMR (Appendix A).

### 2.2. Preparation and Characterization of Nanoreactors

The protocol we developed [31] for making nanoreactors is a simple thin-film hydration method, excluding any additional processing steps, primarily extrusion through nanoporous membranes, homogenization process, freeze−thaw cycles, and the use of organic solvent(s). The method we used, first, avoids the denaturing effect of organic solvents on enzymes, and second, prevents the shear-stress-induced unfolding of enzymes. The size, zeta potential, polydispersity, shape, and morphology of nanoreactors were determined using dynamic light scattering (DLS) and transmission electron microscopy (TEM). The characteristics of empty nanoreactors are presented in Table 1.

The colloidal characteristics of polymers **1a** and **1b** were in agreement with the theory about the hydrophilicity parameter of PEG-PPS polymers [45,46,47]. As determined using DLS, the diameter of nanoreactor **1a** was about 40 nm. The size did not change with increasing **1a** concentration (Appendix A). Thus, **1a** with f_PEG_ = 0.44 formed micelles and **1b** with f_PEG_ = 0.27 organized into polymersomes with a size close to 100 nm. Varying the concentration of **1b** from 0.1 to 1% (*w*/*w*) did not affect the size and polydispersity (PDI = 0.17 ± 0.3) of the polymersomes. The diameter of **1b** polymersomes was determined using the number parameter and intensity at different concentrations (Appendix A). The zeta potential of nanoreactors (ξ, mV) was in the range from −4 ± 1 to −7 ± 0.4 mV. The types of nanoreactor morphology determined using TEM are presented in Figure 2. Empty micellar-type **1a** nanoreactors were spheres with a dark polymeric core (Figure 2A and Appendix A). **1b** nanoreactors looked like empty vesicles with a membrane and an empty core (Figure 2B and Appendix A). **1b** nanoreactors were spherical and uniform, confirming the low PDI. The sizes provided by both methods (DLS and TEM) were very close.

One important parameter of nanoreactors is the membrane permeability for reactants/products. The monitoring of membrane permeability was performed by analyzing the release of p-nitrophenol (pNp), which is the POX hydrolysis product, from nanoreactors. pNp-loaded nanoreactors were prepared using the same method as for empty nanoreactors. The characteristics of pNp-loaded nanoparticles are presented in Table 2.

The characteristics of nanoreactors **1a** and **1b** with pNp loading were different. This confirmed the different morphologies of the nanoreactors: micelles for **1a** and polymersomes for **1b**. For nanoreactors based on **1a**, the size (Z-average, nm) did not change when incorporating pNp, and the encapsulation efficiency (EE, %) was about 80%. The UV spectra of pNp for the determination of *EE* were presented in Appendix A. The nanoreactor size for **1b** increased from 88 ± 0.1 nm to 96 ± 0.6 nm (C_pNp_ = 0.2% *w*/*w*) and from 94 ± 2 nm to 145 ± 1 nm (0.5% *w*/*w*). EE for the **1b** nanoreactors was about 99%. Monitoring the permeability for pNp showed a non-linear, “explosive” release of pNp (Figure 3 and Appendix A). The release of 60% pNp was very fast and a burst release was observed for the control and **1a** and **1b** nanoreactors. Then, the release for the next 40% pNp was slow. The full release (100%) of pNp from **1b** nanoreactors took more than 24 h.

Thus, nanoreactor **1b** with vesicular morphology, high encapsulation efficiency, and loading capacity for hydrophilic compound pNp was selected for the encapsulation of the enzyme.

### 2.3. Enzyme-Loaded Nanoreactor Based on **1b** Polymersomes

The development of enzyme nanoreactors included varying the polymer concentration from 0.1 to 0.5% (*w*/*w*) and varying the enzyme concentration from 2.5 to 12.5 μM (Table 3). On the one hand, our detoxification nanodevice approach suggests the encapsulation of the maximal enzyme concentration for a fast and complete neutralization of POX. On the other hand, there was a risk that only a fraction of the enzyme molecules would be encapsulated into the inner “water core”, while another fraction of enzymes would be partially retained on the polymersomes’ surfaces, forming a “corona” outside. The optimal enzyme concentration was thus determined.

The loading of the enzyme led to an increase in the sizes of the nanoreactors. The diameter (Z-average) varied from 84 (empty nanoreactor) to 113 nm (C**_1b_** = 0.1% *w*/*w*), from 88 (empty nanoreactor) to 114 nm (C**_1b_** = 0.2% *w*/*w*), and from 94 (empty nanoreactor) to 106 nm (C**_1b_** = 0.5% *w*/*w*). The zeta potential slightly shifted to a more negative value. The enzyme-encapsulation efficiency, as determined using UV spectroscopy (Appendix A), was high, i.e., close to 90%, even at the highest enzyme concentrations (12.5 µM). The loading capacity (LC, %) remained high, even with the increased enzyme concentration. The maximum LC (%) was 41 ± 1% (C**_1b_** = 0.2% *w*/*w*).

### 2.4. In Vitro Study of Enzyme Nanoreactors

As mentioned above, E-nRs s work as reactant/product diffusion nanodevices for the neutralization of the organophosphate POX in the bloodstream [30,48]. The POX concentration in blood can reach 6 µM in real field conditions of the most severe cases of poisoning [49]. It is important to work at molar concentrations of enzymes exceeding the POX concentration, both for the possibility of POX self-diffusion into nanodevices and for the reaction of POX with the enzyme (E) inside the reactor under second-order conditions, i.e., [E] > [OP]. This is the main difference between detoxification in enzyme nanoreactors and detoxification by soluble stoichiometric and catalytic bioscavengers working under a pseudo-first-order condition [50].

The concentration of reactants, namely, the enzyme and its substrate, inside the nanoreactor is a mandatory parameter for reactions taking place inside a closed space. This parameter was determined via nanoparticle tracking analysis (NTA) using the total concentration of nanoparticles of 3.21 ± 0.36 × 10^13^ particles/mL [31]. In view of the foregoing, a nanoreactor containing 0.5 mM of enzymes, that is, more than two orders higher than the POX concentration entering inside the nanoreactors, (form. no. 8, Table 3) was chosen for the in vitro study. The main characteristics of nanoreactors, namely, the size and shape of nanoreactors containing the enzyme (form. no. 8), stability in blood plasma, kinetics of POX hydrolysis in the cuvette (in vitro simulation), and nanoreactor permeability for products (pNp) of the POX hydrolysis reaction, are shown in Figure 4 and Appendix A.

The size of nanoreactors was homogenous and close to 100 nm, as confirmed by the low polydispersity index (Table 3). Monitoring the stability of nanoreactors (form. no. 8, Table 3) in mouse blood at 37 °C within 1 h (Figure 4B), and then at all stages of plasma preparation (centrifugation, freezing at −20 °C during 7 days) showed that the enzyme nanoreactors were stable: the diameter (Z-average, nm) did not change significantly (Appendix A) and the PDI slightly increased up to 0.29. The hydrolysis kinetics of POX (at three different concentrations) (Figure 4C) and monitoring the release of product pNp at λ = 400 nm (Figure 4D and Appendix A) indicate that toxic POX was hydrolyzed almost instantly in less than 10 s into harmless products. Thus, POX very rapidly penetrated the nanoreactors, where it was hydrolyzed by the enzyme. For the next in vivo study, the nanoreactor sample (form. no. 8, Table 3) after the purification step, which involved removing the unencapsulated part of the enzyme, was used.

### 2.5. Prophylaxis and Post-Exposure Treatment Trials of Nanoreactors In Vivo

#### 2.5.1. LD_50_ Shifts

POX s.c. administration to mice at doses of 0.5, 0.625, 0.65, 0.7, 0.75, and 1.25 mg/kg caused sedation, labored breathing, tremors, and death due to respiratory failure. The POX LD_50_ was determined to be 0.66 mg/kg (Figure 5, Appendix A). The animals of the control groups that received empty nanoreactors (control 1) and the solvent (saline-ethanol, 10% vol.) (control 2) were slightly depressed immediately after injection, but after 10 min, their initial condition was restored. The prophylactic enzyme-loaded nanoreactor injection shifted the POX LD_50_ from 0.66 mg/kg to 15.62 mg/kg. In the post-exposure treatment using enzyme-loaded nanoreactor administration, the LD_50_ shift was less, but the treatment was still very effective, providing an increase in LD_50_ from 0.66 mg/kg to 5.29 mg/kg.

#### 2.5.2. Behavioral Test

To assess the protective effect of treatments on the behavior of poisoned mice, labyrinth tests were performed. The elevated plus maze test was carried out to measure fear or anxiety. Rodents have a natural tendency to avoid the open arms of the maze and spend more time in the confined closed-arm area. Therefore, the ratio of time spent in the open arm to that of total time (including the time spent at the maze center) (%) was calculated and applied as an index of anxiety. This behavioral test suggests that mice prefer the closed arms of the maze to the open arms [51]. Our results showed that mice spent a significantly longer time in the closed arms, and entered these arms more frequently than in the open arms (Appendix A). On the first day of the experiment, the most marked difference was observed with prophylactically treated animals (the fourth group). They spent less time in the center of the maze (12.7 ± 6.7 s) and engaging in entries into the closed arms (1.4 ± 0.1 s) and rearings (1.3 ± 0.8 s) compared with the first control group (64.00 ± 15.1 s, 6.9 ± 1.7 s, 13.1 ± 1.8 s, respectively; *p* < 0.05) (Appendix A). In addition, a significant increase (*p* < 0.05) of time spent in the closed arms (286.6 ± 6.8 s) compared with the fifth group (200.4 ± 26.5 s) and second control group (160.8 ± 16.8 s) was noted, in addition to a decreased number of head dips (0.7 ± 0.5) and grooming (Appendix A, Appendix A). A reduced time in the center of the maze (the “decision point” for exploration) [52] suggests the animals displayed an anxious nature. The decrease in time spent in the center was associated with an increase in time spent in the closed arms. These data reveal a profile of enhanced anxiety. There was no difference between any groups in the number of entries into the open arms and defecation, as well as the time spent in the open arms and the time spent before entry to the open arms. Since one of POX poisoning symptoms is diarrhea, it was important to find out the absence of excessive defecation in the fourth and fifth groups. This sign also shows the good protective properties of enzyme-loaded nanoreactors against the toxicity of POX (Appendix A). On day 30, the only difference between all groups was the number of head dips between the third (0.4 ± 0.2) and fourth groups (3.7 ± 0.9) (*p* < 0.05) (Appendix A). The absence of difference in the other nine analyzed parameters between groups indicates the recovery of mice after the toxicant exposure.

In comparison, the behavior of the fourth group between the 1st and 30th days shows that animals spent less time in the closed arms (194.8 ± 28.5 s) and reared as often as the other mice (12.8 ± 2.1). That “central platform behavior” when increasing the time in the center of the maze suggests that mice recovered after the toxicant exposure. In the fifth group of mice, the number of entries into the open arms (2.3 ± 1.1 vs. 0.4 ± 0.2) decreased, as well as head dips (9.2 ± 2.9 vs. 2.5 ± 0.6), by the 30th day of the study. This also indicates a decrease in the toxic effects of POX. In the second control group, there was no difference in the studied parameters compared with the first control group, both on the first and last days of the study. This indicates the absence of POX solvent (EtOH) effects on the animal behavior. We only noticed a decreased number of head dips on the 30th day (2.4 ± 0.9) of the test compared with the 1st day (11.4 ± 4.3). There was no difference between the 1st and the 30th day in the first control group and between all groups in the time spent in the open arms and the number of grooming sessions and entries into closed arms.

### 2.6. Pharmacokinetics Study in Mice

Enzyme activity levels in the mouse blood as a function of time after injection were analyzed by measuring the activity of the enzymes toward POX in serum samples. Two groups of animals received intravenous administration of enzyme or E-nRs: the first group was injected with a free enzyme (dose of 3.7 mg/kg) for the first time, while the second group received enzyme-loaded nanoreactors (dose of 3.7 mg/kg) for the first time, and the second injection was 1 month after the first one of the nanoreactors (30 days). Pharmacokinetic profile curves are presented in Figure 6.

Enzyme activity versus time plots from the intravenous route of administration were fitted to the one-compartment pharmacokinetic model (Equation (3)). The pharmacokinetic parameters, including the elimination half-time (t_1/2α_), were calculated and are presented in Table 4.

Statistical analysis of PK profiles was performed using Student’s *t*-test and ANOVA are presented in the Appendix A. The difference between the free enzyme (1 day) and enzyme-loaded nanoreactors (1 day) was submitted to a two-sample *t*-test and was shown to be not significant (Appendix A). The difference between enzyme-loaded nanoreactors (1 day) and enzyme-loaded nanoreactor second injection (30 days) was not significant either (Appendix A). According to the one-way ANOVA, the differences between the groups of mice given the free enzyme (1 day), enzyme-loaded nanoreactors (1 day), and enzyme-loaded nanoreactor second injection (30 days) were not significantly different (Appendix A).

Tuning the membrane envelope structure of enzyme nanoreactors was investigated using three-block copolymer poly(ethylene glycol)-block-poly(propylene sulfide)-block-poly(ethylene glycol) (mPEG–PPS–mPEG). The structure and all characteristics of the mPEG–PPS–mPEG nanoreactors for in vivo detoxification of organophosphates were published in our previous article [31]. Two groups of animals were injected intravenously: the first group received the free enzyme (dose of enzyme 5.76 mg/kg) for the second time one month after the first injection; the second group received the enzyme-loaded nanoreactors with a dose of enzymes of 5.76 mg/kg for the second injection, 1 month after the first injection of nanoreactors (30 days). Pharmacokinetic profile curves and pharmacokinetic parameters are presented in Appendix A. The statistical analysis of the PK profiles was performed using Student’s *t*-test for the free enzymes and nanoreactors made using three-block copolymers showed no significant difference (Appendix A).

### 2.7. Immunological Analyses

Plasma of animals collected after the injection of the free enzymes, solvent, POX, and enzymes was encapsulated into two types of nanoreactors: the first one made of a three-block copolymer (PTE-mPEG–PPS–mPEG) and the second one made of a di-block copolymer (PTE-mPEG–PPS), as well as empty nanoreactors, were analyzed using ELISA and Western blotting.

#### 2.7.1. ELISA

Enzyme therapy is often complicated by immune responses to the enzymes, which block efficacy by neutralizing product activity and causing severe adverse effects in patients [53]. Immunoglobulin G (IgG) is a major effector molecule of the immune response and the main type of antibody found in blood [54]. Therefore, in this work, we used the indirect enzyme-linked immunosorbent assay (ELISA) to analyze the presence of IgG-neutralizing antibodies in mice sera against the PTE enzyme. Conditions for ELISA assays were standardized using checkerboard titrations. We tested the range of concentrations for the secondary antibody from 1:20,000 to 1:40,000 and the following dilution factors for the serum samples: 1:150, 1:200, and 1:250. The highest P/N value of the OD 450 nm ratio between the positive and negative control samples was the final condition of the ELISA. The highest value (4.12) was detected when we used the dilution factor 1:40,000 for the HRP-conjugated antibody and 1:250 for the tested serum samples (Appendix A).

The sensitivity of ELISA was determined using the detection results of PTE antibody-positive serum. The specificities of ELISA were determined using the detection results of PTE antibody-negative serum. According to the receiver operating characteristic (ROC) analysis, the optimal cut-off value was determined as 0.55 (Appendix A). This value was used as the cut-off level for the analysis. Samples were considered positive for IgG against PTE if the OD 450 nm value was greater than 0.6.

The optimized assay was carried out to detect antibodies in mice serum samples (Figure 7). The experiment design is presented in Figure 7A. As we expected, we did not detect the development of a non-specific immune response to PTE after a single injection of empty nanoreactors, solvent, or paraoxon (Figure 7B). We detected PTE-specific IgG in the sera after the second injection of the PTE-mPEG−PPS−mPEG nanoreactor (Figure 7C). The anti-PTE IgG signal levels after the first injection of the nanoreactor were below the cut-off level. According to the literature, murine IgG antibodies appear with a lag period of about 2 days, with a maximum at 10–14 days [55]. Importantly, with the PTE-mPEG−PPS nanoreactor, all tested samples were identified as negative for IgG antibodies, except for the positive control (Figure 7D). Thus, according to the ELISA analysis, we observed the development of the secondary immune response after the second injection with the PTE-mPEG−PPS−mPEG nanoreactor, while an absence of signal was detected with the PTE-mPEG−PPS nanoreactor.

#### 2.7.2. Western Blot

Since PTE-specific IgG was detected in the sera after the second injection of both types of PTE-encapsulated nanoreactors, as determined via ELISA, we wanted to additionally investigate the possible development of the immune response and confirm the obtained results using Western blotting as an alternative method. Free PTE and diluted murine serum samples were used as antigens and primary antibodies, respectively. In the positive control samples, we detected a single ~30 kDa band, while the negative controls exhibited the absence of a signal (Appendix A). This indicates that antibodies from the tested sera specifically bind to the enzyme.

Consistent with ELISA results, we did not detect the presence of IgG antibodies against PTE after a single injection of the empty nanoreactor, free PTE, solvent, or paraoxon (Figure 7E). Similarly, we did not observe the development of the immune response 24 h after a single injection of PTE-mPEG−PPS−mPEG nanoreactors. In contrast, a statistically significant elevation in anti-PTE-IgG levels was observed following the second administration of PTE-mPEG−PPS−mPEG nanoreactors (Figure 7F).

According to the literature, Western blot analysis may give higher sensitivity for the detection of antibodies in host response studies compared with ELISA [56,57]. Interestingly, in contrast with the ELISA results, we observed a signal following two injections, even with the PTE-mPEG−PPS (Figure 7E), although the detected difference was not statistically significant (Figure 7F).

To conclude, we demonstrated the immune response development following two administrations of the PTE-encapsulated nanoreactor using two distinct analytical approaches: ELISA and Western blotting. We detected a two-fold decrease in the antibody amount with PTE-mPEG−PPS nanoreactors compared with the PTE-mPEG−PPS−mPEG nanoreactors. Two important points may explain these results: (a) the concentration of enzyme used in mPEG−PPS nanoreactors was two times lower than in first nanoreactors, thus inducing a lower immune response; (b) the membrane envelope of the mPEG−PPS nanoreactors made using a di-block copolymer without an SS bond were less susceptible to be rapidly destroyed in the bloodstream.

## 3. Materials and Methods

### 3.1. Subsection

#### Chemicals

Ethyl-Paraoxon (POX, purity ≥ 90%) was obtained from Sigma-Aldrich, Canada; p-nitrophenol (pNp) 99% pure was obtained from Alfa Aesar, Karlsruhe, Germany; poly(ethylene glycol) methyl ether (mPEG; average Mn = 2000) was obtained from Sigma-Aldrich (Saint Louis, MO, USA); Propylene Sulfide (PS; stabilized with Butyl Mercaptan) was obtained from Tokyo Chemical Industry Co., Ltd. (Tokyo, Japan); and Potassium thioacetate (98%) was obtained from Sigma-Aldrich (Switzerland). All other chemicals and solvents were of chemical or biochemical grade. Ultra-purified water (18.2 MΩ cm resistivity at 25 °C) was produced from Direct-Q 5 UV equipment (Millipore S.A.S. 67120 Molsheim, France).

IR spectra were recorded on Bruker Tensor-27 and Vector-22 instruments. NMR experiments were carried out on 400 MHz [400.1 MHz (^1^H), 100.6 MHz (^13^C{^1^H})] or 600 MHz [600.1 MHz (^1^H), 150.9 MHz (^13^C{^1^H})] spectrometers equipped with a pulsed gradient unit capable of producing magnetic field pulse gradients in the z-direction of 53.5 G cm^−1^. All spectra were acquired in a 5 mm gradient inverse broadband probe head. Chemical shifts (δ) are expressed in parts per million relative to the residual ^1^H and ^13^C signal of CDCl_3_ and the signals are designated as follows: s, singlet; d, doublet; t, triplet; m, multiplet. Coupling constants (*J*) are in hertz (Hz)

### 3.2. Enzyme Source and Catalytic Activity against POX

The enzyme was described in our previous work [31]. It was a multiple mutant of the hyperthermophilic archaea *Saccharolobus solfataricus* phosphotriesterase (PTE)-like lactonase (PLL) that was functionally expressed in *E. coli* BL21 (DE3) and purified to homogeneity as previously described [31]. The enzyme was a dimer of 70 kDa (*Sso*Pox-IIIC1). Five mutations (L72C/Y97F/Y99F/W263V/I280T) were incorporated via directed evolution to enhance the phosphotriesterase (PTE) activity toward organophosphates (OPs). With paraoxon (POX) as the reference OP, the catalytic behavior was michaelian. At pH 7.4 and 25 °C, K_m_ = 719 ± 118 µM, k_cat_ = 73.5 ± 1.7 s^−1^, and k_cat_/K_m_ = 1.02 ± 0.25 × 10^5^ M^−1^s^−1^ [31]. This mutant maintained a high thermostability (T_m_ = 96.3 °C), ensuring compatibility for encapsulation methods and long-term stability [58,59]. The purified enzyme was lyophilized and stored at −20 °C.

### 3.3. Synthesis Polymers for Nanoreactor Construction

For the synthesis of the block copolymers **1a** and **1b**, the *one-pot* method described by Napoli [60] was used based on commercially available polyethylene glycol monomethyl ether tosylate (MPEG-2000-Tos). Next, mPEG-2000-Tos tosylate was treated with an excess of potassium thioacetate (AcSK) in a DMF medium to obtain mPEG-2000-SAc thioacetate [61]. Next, mPEG-2000-SAc was treated with 1.1 equivalents of sodium methoxide in THF, followed by the addition of propylene sulfide (40 and 80 equivalents, respectively, for polymers **1a** and **1b**). After 45 min under an argon atmosphere, the polymerization was stopped by adding 5 equivalents of benzyl bromide and left to mix overnight to obtain polymers 1a and 1b. To isolate block copolymers 1–2, the solvents were removed on a rotary evaporator, dried in a vacuum for 6 h, and washed with petroleum ether three times to remove the unreacted benzyl bromide and nonpolar oligo/polymers of polypropylene sulfide that did not contain fragments of methoxypolyethylene glycol, which were formed in a small amount during polymerization. Subsequently, the block copolymers were additionally dried in a vacuum for 1 h and were obtained as light brown oils. The ratio of the units was determined by integrating the signals in the ^1^H NMR spectra, using a singlet of the terminal methoxy group as a reference point, which was clearly distinguishable and manifested separately from other signals. Integration of the signal of the methylene group, which appeared as a multiplet with a shift of about 3.8 ppm, allows for controlling the benzylation completeness of the terminal sulfhydryl group.

mPEGSAc-2000 ^1^H NMR (600 MHz, CDCl_3_) δ: 3.59–3.52 (s, 200H, CH_2_ broad, PEG chain protons), 3.47–3.44 (m, 2H, -OCH_2_CH_2_S-), 3.28 (s, 3H, CH_3_O), 3.00 (t, *J* = 6.4 Hz, 2H, -CH_2_SCOCH_3_), 2.24 (s, 3H, CH_3_C(O)S) ppm [39].

mPEG_44_-PPS_33_-Bn (**1a**) ^1^H NMR (400 MHz, CDCl_3_) δ: 7.36–7.32 (m, 3H, PhH), 7.26–7.23 (m, 2H, PhH), 3.86–3.77 (m, 2H, CH_2_Ph), 3.73–3.63 (brm, 200H, CH_2_ (PEG)), 3.55 (m, 2H, OCH_2_CH_2_S), 3.40 (s, 3H, OCH_3_), 3.02–2.85 (m, 66H, CH_2_ (PPS)), 2.72–2.59 (m, 33H, CH (PPS)), 1.45–1.28 (m, 99H, CH_3_ (PPS)) ppm. ^13^C NMR (101 MHz, CDCl_3_) δ: 128.82 (s, m-Ph), 128.59 (s, *o*-Ph), 127.09 (s, *p*-Ph), 71.91 (s, OCH_2_CH_2_S), 71.08 (s, CH_2_OCH_3_), 70.60 (s, CH_2_ (PEG)), 70.55 (s, CH_2_ (PEG)), 70.49 (s, CH_2_ (PEG)), 70.37 (s, CH_2_ (PEG)), 59.06 (s, OCH_3_), 41.32 (s, CH (PPS)), 41.31 (s, CH (PPS)), 41.22 (s, CH (PPS)), 38.46 (s, CH_2_ (PPS)), 38.41 (s, CH_2_ (PPS)), 38.38 (s, CH_2_ (PPS)), 38.35 (s, CH_2_ (PPS)), 35.53 (s, CH_2_Ph), 32.42 (s, OCH_2_CH_2_S), 20.81 (s, CH_3_ (PPS)), 20.78 (s, CH_3_ (PPS)) ppm. FT-IR 3419, 2957, 2883, 2739, 2695, 1467, 1453, 1415, 1373, 1360, 1343, 1280, 1234, 1174, 1148, 1111, 1061, 1003, 963, 947, 843, 734, 690, 531 cm^−1^.

mPEG_44_-PPS_70_-Bn (**1b**) ^1^H NMR (400 MHz, CDCl_3_) δ 7.35–7.29 (m, 3H, PhH), 7.26–7.22 (m, 2H, PhH), 3.79 (m, 2H, CH_2_Ph), 3.67–3.61 (brm, ~200H, CH_2_ (PEG)), 3.55 (m, 2H, OCH_2_CH2S), 3.38 (s, 3H, OCH_3_), 2.96–2.84 (m, ~140H, CH_2_ (PPS)), 2.68–2.58 (m, ~70H, CH (PPS)), 1.46–1.29 (m, ~210H, CH_3_(PPS)) ppm. ^13^C NMR (101 MHz, CDCl_3_) δ: 138.22 (s, i-Ph), 128.74 (s, *m*-Ph), 128.49 (s, *o*-Ph), 126.99 (s, *p*-Ph), 71.84 (s, OCH_2_CH_2_S), 71.04 (s, CH_2_OCH_3_)), 70.40 (s, CH_2_ (PEG)), 58.94 (s, OCH_3_), 41.25 (s, CH (PPS)), 41.18 (s, CH (PPS)), 41.14 (s, CH (PPS)), 38.41 (s, CH_2_ (PPS)), 38.35 (s, CH_2_ (PPS)), 38.32 (s, CH_2_ (PPS)), 35.47 (s, CH_2_Ph), 32.38 (s, OCH_2_CH_2_S), 20.72 (s, CH_3_ (PPS)) ppm. FT-IR 3454, 2958, 2889, 2740, 2695, 1466, 1452, 1415, 1373, 1360, 1343, 1307, 1280, 1241, 1175, 1148, 1107, 1061, 1003, 963, 948, 843, 734, 691, 531 cm^−1^.

### 3.4. Procedure of Nanoreactor Preparation

Polymers **1a** and **1b** (0.1–2% *w*/*w*) were dissolved in 1 mL of organic solution ethanol:chloroform (1:1). The homogeneous polymer solution was kept for 24 h for alcohol evaporation. Then, 10 mM Tris-buffer (pH 7.4) was pre-heated to 37 °C and added to rehydrate the polymers at 37 °C in the absence or presence of enzymes and p-nitrophenol (pNp, 0.1% *w*/*w*). The solution was stirred under magnetic stirring (700 rpm) (Heidolph, Germany) for 6 h at 37 °C and then for 24 h at 25 °C.

### 3.5. Characterization of Nanoreactors

#### 3.5.1. The Size, Surface Charge, and Morphology

The mean particle size, zeta potential, and polydispersity index were determined with dynamic light scattering (DLS) using a Malvern Instrument Zetasizer Nano (Worcestershire, UK). The size (hydrodynamic diameter, nm) was calculated according to the Einstein–Stokes relationship *D* = *k_B_T*/3*πηx*, in which *D* is the diffusion coefficient, *k_B_* is the Boltzmann’s constant, *T* is the absolute temperature, *η* is the viscosity, and *x* is the average hydrodynamic diameter of the nanoparticles. The diffusion coefficient was determined at least in triplicate for each sample. The average error of measurements was approximately 10%.

Transmission electron microscopy (TEM) was used to image the size and to reveal the morphology of both empty and enzyme-loaded nanoreactors. TEM images were obtained using a Hitachi HT7700 Exalens microscope (Japan). The images were acquired at an accelerating voltage of 100 keV. Samples (C = 20 μg/mL) were added to 300-mesh copper grids with continuous carbon formvar support films.

#### 3.5.2. Encapsulation Efficiency (EE, %) and Loading Capacity (LC, %)

*EE* (%) and *LC* (%) were assessed for samples containing enzyme and pNp. These parameters were determined indirectly via filtration/centrifugation, measuring the free enzymes and pNp (non-encapsulated) using spectrophotometry. A 100 µL volume of each PTE-loaded polymersome was placed in Amicon^®^ Ultra-4 Centrifugal Filter Ultracel^®^—100K centrifugal filter devices (Millipore Merck KGaA, Darmstadt, Germany) to separate the polymeric nanoreactors and aqueous phases and centrifuged at 3000 rpm and 4 °C for 5 min using a Rotanta centrifuge (Hettich Zentrifugen, Germany). The concentration of free enzyme in the Tris buffer supplemented with Co^++^ = 0.2 mM was quantified via UV absorbance using a PerkinElmer λ_35_ (PerkinElmer Instruments, Waltham, MA, USA) at 265 nm (ε = 93,333 M^−1^ cm^−1^ in 10 mM Tris buffer, Co^++^ = 0.2 mM, pH = 7.4). UV absorbance spectra and calibration curves are presented in the Appendix A. A 100 µL volume of each pNp-loaded nanoreactor was placed in a Nanosep centrifugal filter device 3K Omega (Pall Corporation, New York, NY, USA) to separate the polymeric nanoreactors and aqueous phases and centrifuged at 10,000 rpm for 15 min using a centrifuge MiniSpin plus (Eppendorf AG, Hamburg, Germany). Free pNp was quantified via UV absorbance using PerkinElmer λ_35_ (PerkinElmer Instruments, USA) at 400 nm (ε = 11,554 M^−1^ cm^−1^ in 10 mM Tris buffer pH = 7.4). UV absorbance spectra and calibration curves are presented (in the [31] Appendix A).

The encapsulation parameters *EE*% and *LC*% were calculated against appropriate calibration curves using the following equations [31]:(1)$$EE\left(\%\right)=\frac{Total\;amount\;of\;enzyme-Free\;enzyme}{Total\;amount\;of\;enzyme}\times 100\%,$$
(2)$$LC\left(\%\right)=\frac{Total\;amount\;of\;enzyme-Free\;enzyme}{Total\;amount\;of\;copolymer}\times 100\%,$$

#### 3.5.3. In Vitro Stability of Nanoreactors in Blood

A total of 300 μL (free enzyme and enzyme-loaded nanoreactors) was added to 300 μL of pure blood and incubated for 1 h at 37 °C. Then, the blood was incubated for 1 h at 4 °C and centrifuged for 15 min 2500 rpm at 4 °C. The resulting supernatant was collected and freezed at −20 °C. The free enzyme solution was used as a control. At each step, samples were diluted 100 times, and then the size, zeta potential, and PDI of the nanoreactors were measured over time via DLS using a Malvern Instrument Zetasizer Nano (Worcestershire, UK).

#### 3.5.4. Purification of Enzyme-Loaded Nanoreactors

To remove unencapsulated free enzymes from the PTE-loaded polymersomes, the two enzyme fractions were separated via filtration/centrifugation using an Amicon^®^ Ultra-4 Centrifugal Filter Ultracel^®^—100K centrifugal filter device (Millipore Merck KGaA, Darmstadt, Germany). Fractions of 0.5 mL were centrifuged at 3000 rpm and 4 °C for 5 min using a Rotanta 460 centrifuge (Hettich Zentrifugen, Germany) and monitored with a UV spectrometer at 265 nm. These conditions were determined by monitoring the transmittance of empty nanoreactors under centrifugation conditions over time. The transmittance of empty nanoreactors is presented using UV absorbance spectra and the calibration curve is presented in the Appendix A.

#### 3.5.5. In Vitro the Nanoreactor Permeability

Monitoring of pNp release from nanoreactors was performed using the dialysis bag diffusion method. Dialysis bags retain pNp-loaded nanoreactors and allow the released pNp to diffuse into the medium. The bags were soaked in Milli-Q water for 12 h before use. A total of 0.5 mL pNp-loaded nanoreactors were poured into the dialysis bag. The two bag ends were sealed with clamps. The bags were then placed in a vessel containing 100 mL of 10 mM Tris buffer, pH 7.4, as the receiving phase. The vessel was placed in a thermostatic shaker (New Brunswick, NJ, USA) at 37 °C under a stirring rate of 150 rpm. At predetermined time intervals, 0.5 mL samples were withdrawn and their absorbance at 400 nm was measured using a PerkinElmer λ_35_ spectrophotometer (PerkinElmer Instruments, USA).

### 3.6. In Vitro Simulations of Enzyme-Loaded Nanoreactor Reactions against POX in Spectrophotometer Cuvette

Nanoreactor simulation of POX inactivation was performed under second-order conditions in 1 cm spectrophotometric cuvettes in 10 mM Tris buffer, pH 7.4, at 25 °C. Enzyme-catalyzed hydrolysis of POX was monitored using the absorbance increase at 400 nm due to the release of its leaving group, namely, pNp; the kinetics of neutralization of POX (5 μM) using the stoichiometric concentrations of PTE (5 μM) was carried out by adding the whole dose of POX in a single volume. The maximum POX concentration, namely, 5 μM, was chosen because it was of the order of the maximum OP concentration determined in human blood in the most severe cases of poisoning by POX. Free enzyme solution under the same conditions was used as a control. In addition, the control of pNp spectra was monitored in the same conditions using dialysis bags containing pNp-loaded nanoreactors and a solution of free enzyme.

### 3.7. In Vivo Study of Free Enzyme and Enzyme-Loaded Nanoreactors

#### 3.7.1. Animals

Male CD-1 mice (weighing 18–22 g) were purchased from a Russian-certified nursery (G. Zhakovich Co, Lietnii Otduh, Moscow region, Russia). Animals were acclimatized for 2 weeks prior to the experiments. They were housed in sawdust-lined polypropylene cages and maintained under standard conditions (12 h light/dark cycle, 22 ± 3 °C, and 50 ± 20% relative humidity). Animals were given a standard pellet diet with free access to food and water. All experimental procedures with animals were performed in accordance with the Ethical Principles in Animal Research and were approved by the Local Ethics Committee of the Kazan Federal University (protocol No. 40).

#### 3.7.2. POX LD_50_ Shifts at Pre- and Post-Exposure Treatments in Mice

Mice were stratified by weight and randomly assigned into groups of three or six per group. POX was extemporaneously diluted in hydroalcoholic isotonic saline solution (EtOH 10% in sodium chloride 0.9%). The final EtOH concentration per dose was 1 mg/kg. The POX LD_50_ was determined using subcutaneous (s.c.) injections at POX doses ranging from 0.5 to 1.25 mg/kg. Injections of 0.2 mL POX solution per 20 g animal were performed s.c. using an insulin syringe. The EtOH 10% in sodium chloride 0.9% (s.c.) and empty nanoreactor (i.v.) solutions were checked in control group 1 and control group 2, respectively. Then, LD_50_ determinations were performed after pre-treatment (prophylactic) and post-exposure (therapeutic) treatment of animals using an enzyme-loaded nanoreactor solution. A single dose (3.7 mg/kg) was injected in a tail vein using an insulin syringe. In pre-treatment LD_50_ shift experiments, the enzyme nanoreactor solution was administered via an injection into the tail vein 5 min before the POX challenge. The prophylactic LD_50_ shift was determined using POX doses ranging from 10 to 25 mg/kg s.c. In the post-exposure treatment trials, the enzyme-containing nanoreactor solution was injected 1 min after the POX challenge at doses ranging from 2.5 to 10 mg/kg s.c. The initial POX doses were selected as doses expected to produce mortality in some animals. Further groups of animals were dosed at higher or lower fixed doses, depending on the mortality in the challenged animal groups until the study objective was achieved. For each dose, 3 animals were used to minimize the number of animals. If in a group of 3 animals, an unequivocal response was obtained (all animals died or survived), then we proceeded to the next dose.

All animals were observed individually for symptoms and mortality after dosing with special attention during the first 4 h and twice a day thereafter for two weeks. Poisoned animals that did not survive died in less than 24 h. Dead animals were autopsied. The LD_50_ was calculated via Probit analysis using IBM SPSS Statistics software, Version 27.0.0.

#### 3.7.3. Elevated plus Maze Test

To assess the prophylactic and post-exposure treatment effects of enzyme-loaded nanoreactors, animal behavior was examined on the 1st day and after 30 days using an elevated cross maze for mice (RPC OpenScience Ltd., Moscow region, Krasnogorsk, Russia). Fifteen minutes after the injection, each mouse was placed on the junction of the open and closed arms of an elevated cross maze, facing the open arm. Video filming over 5 min was carried out using a digital video system DMK 23GVO24 (Imaging source, Germany). Video decoding was carried out manually [62]. The following parameters were analyzed: time spent on the central junction and before entry to closed arms; number and duration of entries to open and closed arms; and numbers of defecation, grooming, head dips, and rearings. Five groups of animals were formed

In the 1st control group, instead of POX and enzyme-loaded nanoreactors, equal volumes of sterile saline solution were s.c. and i.v. injected into mice. For prophylactic experiments, the i.v. injection of saline solution (100 μL/20 g animal weight) was administered to five animals in this group, and then, 5 min later, saline solution (200 μL/20 g animal weight) was s.c. injected. For the control of the post-exposure treatment, an s.c. injection of the same volume of saline solution was given to five other mice in this group, and then after 1 min, an i.v. saline solution was administered.

In the 2nd POX solvent control group, instead of POX and enzyme-loaded nanoreactors, equal volumes of POX solvent (EtOH 10% in sodium chloride 0.9%, s.c.) and sterile saline solution (i.v.) were administered to the mice. For the prophylactic experiments imitation, the i.v. injection of saline solution was administered to three animals, and 5 min later, the solvent was s.c. injected. For post-exposure treatment, the s.c. injection of solvent was given to two other mice, and then, after 1 min, an i.v. saline solution was administered.

In the 3rd empty nanoreactors control group, instead of POX and enzyme-loaded nanoreactors, equal volumes of sterile saline solution (s.c.) and empty nanoreactors solution (i.v.) were administered to the mice. For the prophylactic experiments, i.v. injection of empty nanoreactor solution was administered to three animals, and 5 min later, a saline solution was s.c. injected. To perform post-exposure treatment experiments, an s.c. injection of saline solution was given to two other mice, and then, 1 min later, an i.v. empty nanoreactor solution was administered.

In the 4th prophylactic experiment group, enzyme-loaded nanoreactors were i.v. injected, and 5 min later, POX at 7.8 mg/kg (i.e., pretreatment determined for half the LD_50_) was s.c. injected into the animals.

In the 5th post-exposure treatment experimental group, enzyme-loaded nanoreactors were i.v. injected 1 min after a POX s.c. challenge at 2.6 mg/kg (i.e., post-treatment determined for 1/2 LD_50_) to the mice.

Data were statistically analyzed using ANOVA, with the significance level set at *p* < 0.05.

### 3.8. Pharmacokinetics in Mice

The recommended maximum volume for i.v. administration in mice is 0.1 mL. Proceeding from this recommended volume, free enzyme solutions and enzyme-loaded nanoreactor solutions (enzyme dose of 3.7 mg/kg) were slowly administered into the tail vein of mice weighing 20–30 g.

Two groups of animals were set: enzyme solutions and enzyme-loaded nanoreactor solutions were injected into the tail vein of each group. For each predetermined time interval (10 and 30 min and 1, 2, 4, and 24 h after injection) in each group, blood samples were collected into test tubes. After 1 month, the same dose of enzyme-loaded nanoreactor solutions (3.7 mg/kg) was injected into the tail vein of the second group into the same mouse, and blood samples were collected at the same time intervals (10 and 30 min and 1, 2, 4, and 24 h after injection). After the collection of CD-1 mice blood, blood was incubated for 1 h at 37 °C; then, it was kept for 1 h at 4 °C, centrifuged for 15 min under 2500 rpm at 4 °C, and the serum supernatant was collected and frozen at −20 °C. Phosphotriesterase activity with POX as the substrate was subsequently assayed in each sample.

#### 3.8.1. Assay of Enzyme Activity in Serum for Pharmacokinetic Analysis

A total of 15 μL of serum was added in 1 cm path spectrophotometric cuvettes containing Tris buffer (10 mM, pH 7.4, supplemented with 0.2 mM CoCl_2_). The enzyme activity was determined under standard conditions at 25 °C, the POX stock solutions (0.1 M) were in ethanol (EtOH), and the final EtOH in the cuvette was 1% (vol.). Steady-state kinetics was recorded by monitoring the release of pNp at 400 nm for 120 s.

Pharmacokinetic data were analyzed using the simple one-compartment open model for intravenous bolus injection [63] using Equation (3):(3)$$E_{t\;}=E_{0}\;e^{-\alpha t}$$
where *E_t_* is the enzyme activity in plasma at time *t* after the flash injection, *E*_0_ is the extrapolated initial value of enzyme activity in plasma at *t*_o_, and α is the rate constant of enzyme elimination from the blood compartment. The half-time of pharmacokinetics is
(4)$$t_{1/2}=ln2/\alpha$$

#### 3.8.2. Statistical Analysis

Pharmacokinetic profiles between the free enzyme and the enzyme-loaded nanoreactors 1st injection (1 day) and 2nd injection (30 days) were compared via a one-way ANOVA multiple comparisons test. Pharmacokinetic parameters between the free enzyme and enzyme-loaded nanoreactors 1st injection and enzyme-loaded nanoreactors 1st injection (1 day) and 2nd injection (30 days) were compared via individual unpaired Student’s *t*-tests.

### 3.9. Immune Response to Injected Enzyme-Loaded Nanoreactors

Because our main concern was a possible immune response to PTE with iatrogenic effects after the injection of free enzyme or enzyme nanoreactors, we checked the immune response to PTE by ELISA and Western blotting.

#### 3.9.1. Enzyme-Linked Immunosorbent Assay (ELISA)

Maxibinding polystyrene flat-bottom microplates (SPL Life Sciences, Gyeonggi-do, Republic of Korea) were coated with 100 ng of *Saccharolobus solfataricus* phosphotriesterase protein per well in a carbonate-bicarbonate buffer (pH 9.2) and incubated at 4 °C overnight. All following steps were performed at room temperature unless specified otherwise. Wells were washed 3 times with 300 µL PBS-T and blocked with 5% (*w*/*v*) skimmed milk (Vamin, Russia) for 120 min.

The optimal concentrations of secondary antibody and serum sample dilution were determined via the checkerboard titration method using positive and negative control samples. The optimal ELISA reagent concentration was assumed to be for those showing the highest discrimination between the positive and negative sera. After optimization of the assay, 11 serum samples were analyzed using indirect ELISA. A pool of three positive and three negative serum samples was included in each ELISA plate to monitor the accuracy of the assay.

Both controls and the serum samples were diluted 250-fold in PBS-T with 1% bovine serum albumin (BSA, VWR Life Science, Radnor, PA, USA) and incubated for 1 h at 37 °C. Wells were washed 5 times with PBS-T (300 µL) before adding 100 µL of conjugate solution (1:40,000) in PBS-T containing 1% BSA (anti-Human IgG-HRP; Abcam, UK). After incubating the microplate for 1 h at 37 °C, wells were washed 5 times, as described above, and TMB substrate solution (Xema, Russia) was added (100 µL/well). The reaction was stopped after 10 min in a dark place at 37 °C via the addition of 1 M H_2_SO_4_ and the absorbance was recorded at 450 nm using a Tecan M200 Pro microplate reader (Tecan Life Sciences, Männedorf, Switzerland). The cut-off value was determined using receiver operating characteristics (ROCs) to obtain the best sensitivity and specificity. ROC analysis was performed using GraphPad Prism version 9.3.1 (Graph Pad Software, San Diego, CA, USA).

#### 3.9.2. Western Blot Analysis

A total of 1 µg of PTE enzyme in RIPA buffer containing protease inhibitor cocktail (PIC) (Thermo Fisher Scientific, Waltham, MA, USA) was mixed with 6x loading buffer and loaded into each well of a polyacrylamide gel (4% stacking and 12% separating gel). After electrophoresis, proteins were electroblotted in a semi-dry transfer from gel to polyvinylidene difluoride (PVDF) membrane (Bio-Rad Laboratories, Hercules, CA, USA) according to the BioRad standard protocol. The transfer accuracy was checked via gel staining with Ponceau S (Sigma-Aldrich, Saint Louis, MO, USA). The membrane was cut into separate strips containing one lane each. The blocking of non-specific binding was achieved by placing the membrane overnight in a 5% solution of dry milk (Vamin, Kazan, Russia) in PBST buffer. Further, for IgG antibody detection, membranes were incubated overnight in murine sera diluted with a ratio of 1:300 in PBST buffer containing 2.5% dry milk. Further membranes were exposed to horseradish peroxidase (HRP)-conjugated secondary antibodies (Anti-mouse IgG cat. no. A16078; Thermo Fisher Scientific, Waltham, MA, USA). The visualization of labeled proteins of interest was achieved after placing the membranes in a BioRad HRP-substrate solution for several minutes.

## 4. Conclusions

Nanoreactors based on the PEG-PPS block copolymers with different morphologies were prepared. Micellar nanoreactors were characterized by f_PEG_ = 0.44 and a small size close to 40 nm. For the polymersome nanoreactors, f_PEG_ = 0.27 and the size was about 100 nm, meaning they were monodisperse. For nanoreactors of both morphologies, a fast release of the hydrolysis product pNp was observed at the initial stage and slow at the final stage. Upon receipt of the enzyme reactor, an optimal formulation was obtained with a high enzyme concentration, a high efficiency of 82 ± 7%, and a high enzyme loading of 15 ± 2%. An in vitro simulation of the enzyme nanoreactor operational activity showed its high efficiency in detoxifying 5 µM POX within 10 s. An in vivo study on mice confirmed the high efficacy of this nanodevice in terms of prophylactic and post-exposure therapeutic actions. The LD_50_ shifts were 23.7 and 8, respectively, without any additional drugs currently used for the therapy of OP poisoning. Pharmacokinetic profiles and preliminary immunological studies of the injected enzyme showed no differences between all groups studied but suggested that a fraction of enzyme molecule was partially encapsulated on the nanoreactor outer surface. The elevated plus maze behavioral test indicated the recovery of protected/treated mice after the toxicant exposure within 30 days with no apparent sequelae.

Finally, enzyme nanoreactors containing PTE (PPL), which is effective against POX poisoning, are expected to be effective against poisoning by other OPs used in agriculture. Moreover, nanoreactors containing *Sso*Pox variants may be of utmost interest for protection and emergency treatment against chemical warfare nerve agent (CWNA) toxicity. Indeed, as we mentioned in the Introduction, the *Sso*Pox variant considered in this study was recently shown to efficiently degrade CWNA surrogates, while previously reported variants were shown to be able to degrade chemical warfare nerve agents, such as sarin, cyclosarin, tabun, and soman. The encapsulation of several types of enzymes must also expand the spectrum of molecules to be detoxified. As such, the ultimate goal of this approach is to make safe and effective polyfunctional enzyme nanoreactors.

## Figures and Tables

**Figure 1 ijms-24-15756-f001:**
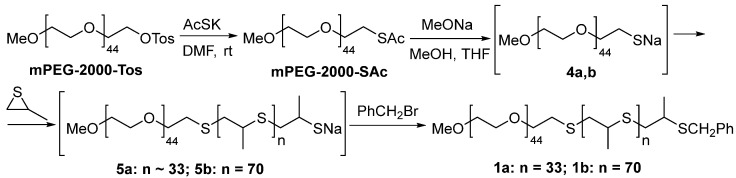
Synthesis of PEG-PPS di-block copolymers.

**Figure 2 ijms-24-15756-f002:**
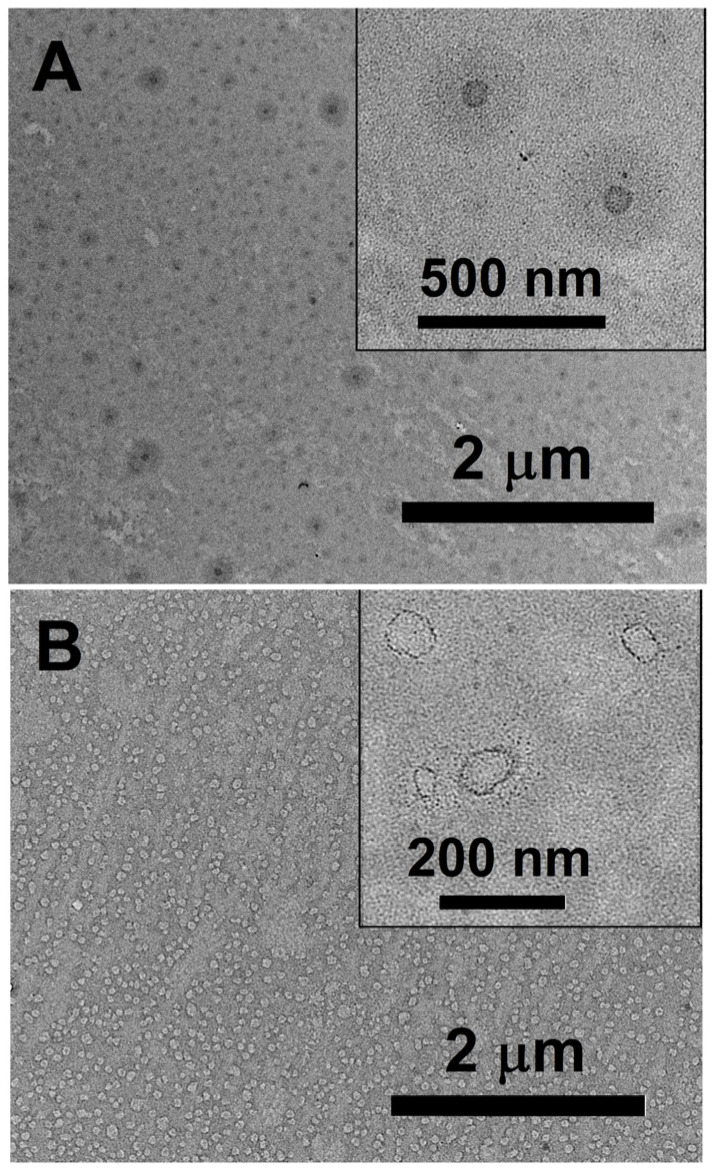
TEM imaging of empty nanoreactors prepared using **1a** (**A**) and **1b** (**B**), C_polymers_ = 10 μg/mL, Tris-Buffer, pH = 7.4, 25 °C. Scale bars are 2 μm for the main (**A**) and (**B**) images, with 500 nm and 200 nm for the insets, respectively.

**Figure 3 ijms-24-15756-f003:**
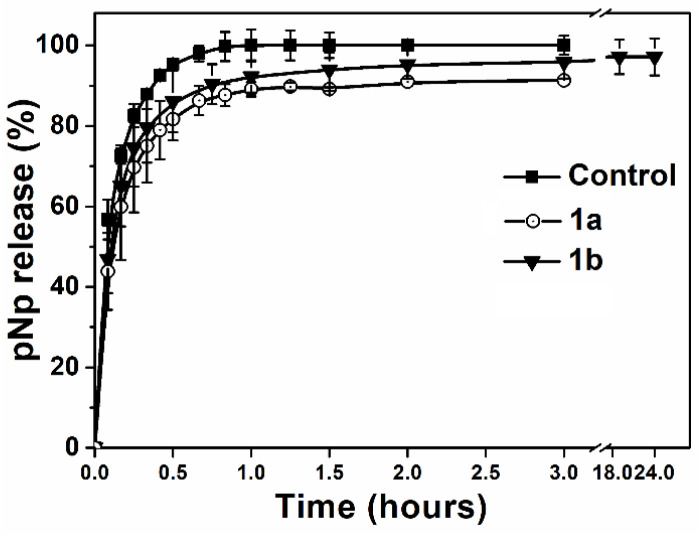
Release of pNp from **1a** and **1b** nanoreactors, where control was the absence of nanoreactors, C_1a_ = C_1b_ = 0.5% (*w*/*w*), C_pNp_ = 0.1% (*w*/*w*), 37 °C, 10 mM Tris-Buffer, pH = 7.4.

**Figure 4 ijms-24-15756-f004:**
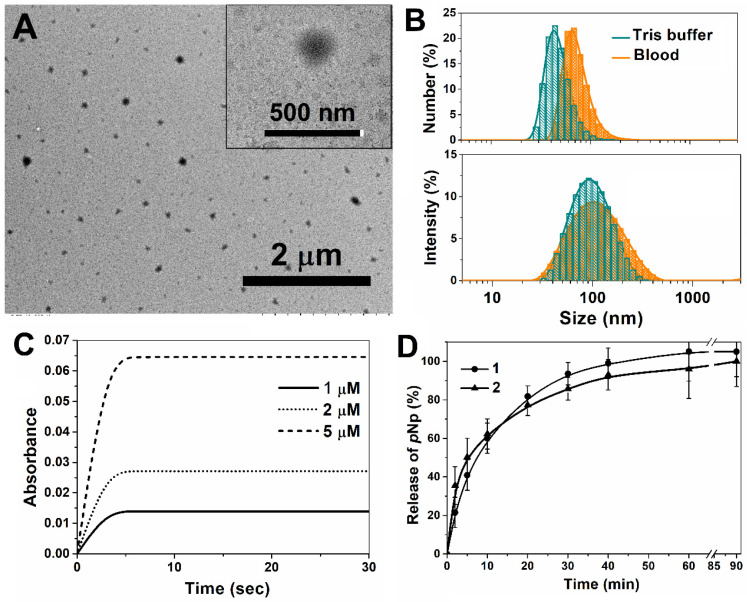
TEM imaging (**A**); size distribution as determined using DLS for the monitoring of stability in vitro conditions in Tris buffer and within 1 h at 37 °C (**B**); in vitro kinetics of POX detoxification process at λ= 400 nm (**C**) under second-order conditions: [E] = 0.5 mM and [POX] = 1–5 µM; pNp release (**D**) of enzyme-loaded nanoreactors **1b**, Tris-Buffer, pH = 7.4, 25 °C.

**Figure 5 ijms-24-15756-f005:**
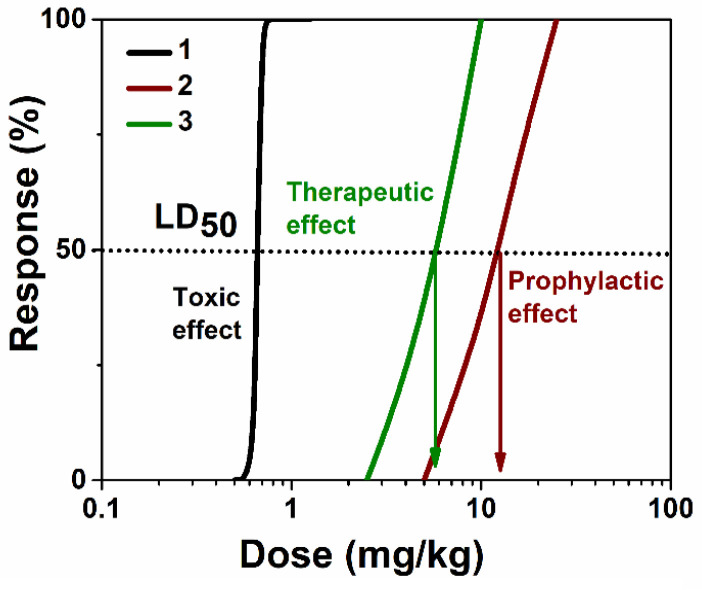
LD_50_ shifts for acute toxicity of POX s.c., where 1 (reference LD_50_)—non-treated animals, 2—animals under prophylaxis, and 3—post-exposure treated animals.

**Figure 6 ijms-24-15756-f006:**
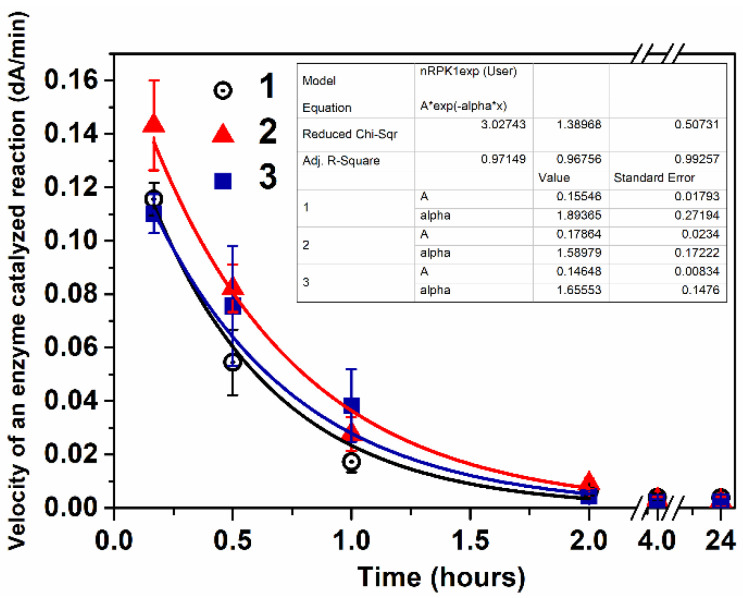
Rate of enzyme-catalyzed reaction of POX hydrolysis versus time in mouse plasma after intravenous injection of free enzymes (1) and enzyme-loaded nanoreactors (2, 3), where (2) is the first injection (1 day) and (3) is the second injection (30 days after the first injection). The dose of enzyme was 3.7 mg/kg. Each point represents the mean ± SD in 6 mice.

**Figure 7 ijms-24-15756-f007:**
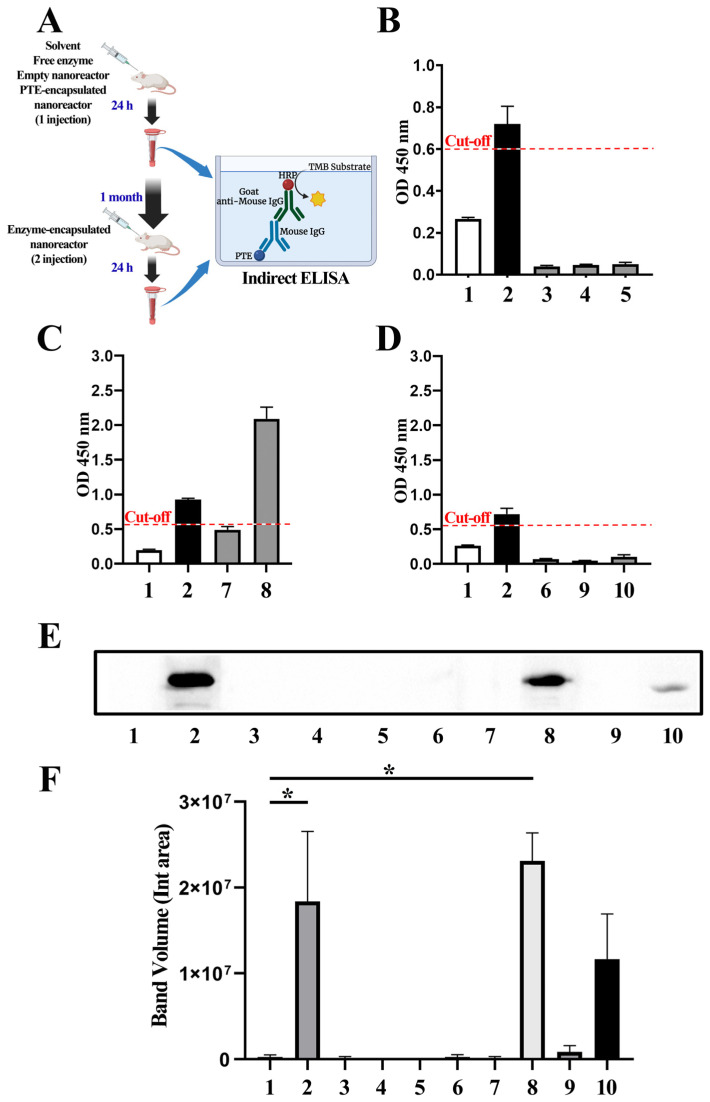
Development of immune response to free or encapsulated PTE enzyme: 1—negative control, 2—positive control, 3—POX, 4—empty nanoreactor, 5—solvent, 6—PTE, 7—PTE-mPEG−PPS−mPEG nanoreactor (1 injection), 8—PTE-mPEG−PPS−mPEG nanoreactor (2 injections), 9—PTE-mPEG−PPS nanoreactor (1 injection), 10—PTE-mPEG−PPS nanoreactor (2 injections). (**A**) Serum samples were collected and prepared for indirect ELISA assay to detect IgG against the free and encapsulated enzyme; (**B**) OD 450 nm values in control and tested serum samples after single paraoxon, empty nanoreactor, or solvent injection; (**C**) OD 450 nm values in control and tested serum samples after first and second injections of PTE-mPEG−PPS−mPEG nanoreactor; (**D**) OD450 nm values in control and tested serum samples after first and second injections of PTE-mPEG−PPS nanoreactor; (**E**) a representative image of the Western blot results for anti-PTE-IgG from 3 independent experiments; (**F**) the levels of the anti-PTE-IgG in murine serum samples were analyzed by densitometry; (*n* = 3; *—*p* ≤ 0.05).

**Table 1 ijms-24-15756-t001:** Empty nanoreactor characteristics, where size is hydrodynamic diameter, Z-average is the mean size, PDI is polydispersity index, and ξ or zeta potential is electrokinetic potential. The medium was 10 mM Tris-Buffer, pH = 7.4, 25 °C.

Polymer	C (%, *w*/*w*)	Size (nm)		Z-Average (nm)	PDI	ξ (mV)
		Intensity	Number			
**1a**	0.5	38 ± 6	21 ± 5	37 ± 1	0.14 ± 0.01	−5.0 ± 1
	1	44 ± 7	24 ± 6	41 ± 0.2	0.12 ± 0.01	−5.5 ± 1
	2	38 ± 6	21 ± 5	38 ± 1	0.15 ±0.02	−5.0 ± 1
**1b**	0.1	91 ± 11	44 ± 10	84 ± 0.3	0.17 ± 0.01	−6 ± 0.3
	0.2	79 ± 11	44 ± 11	88 ± 0.1	0.14 ± 0.01	−7 ± 0.4
	0.5	106 ± 13	44 ± 10	94 ± 2	0.17 ± 0.01	−6.5 ± 0.3
	0.75	122 ± 13	51 ± 11	112 ± 1	0.2 ± 0.01	−5 ± 1
	1	106 ± 13	51 ± 12	106 ± 1	0.18 ± 0.01	−4 ± 1

**Table 2 ijms-24-15756-t002:** Characteristics of pNp-loaded nanoreactors, where size is hydrodynamic diameter, Z-average is the mean size, PDI is polydispersity index, ξ or zeta potential is electrokinetic potential, C_pNp_ = 0.1% (*w*/*w*), 10 mM Tris-Buffer, pH = 7.4, 25 °C.

Polymer	C (%, *w*/*w*)	Size (nm)		Z-Aver (nm)	PDI	ξ (mV)	EE, %	LC, %
		Intensity	Number					
**1a**	0.5	38 ± 5	21 ± 5	37 ± 0.5	0.1 ± 0.01	−8 ± 1	83 ± 4	16.7 ± 0.7
	1	34 ± 9	23 ± 4	33 ± 0.5	0.1± 0.01	-	82 ± 4	8.2 ± 0.4
	2	36 ± 12	23 ± 4	33 ± 0.5	0.12± 0.01	−9 ± 2	96.8 ± 0.2	4.8 ± 0.0.01
**1b**	0.2	106 ± 11	38 ± 8	96 ± 0.6	0.22 ± 0.01	−5.5 ± 0.6	99.9 ± 0.01	49.98 ± 0.1
	0.5	190 ± 21	44 ± 3	145 ± 1	0.23 ± 0.01	−5.2 ± 0.3	99.8 ± 0.1	19.98 ± 0.1

**Table 3 ijms-24-15756-t003:** Enzyme-loaded nanoreactor characteristics, where size is hydrodynamic diameter, Z-average is the mean size, PDI is polydispersity index, ξ or zeta potential is electrokinetic potential, polymer **1b**, 10 mM Tris-Buffer, pH = 7.4, 25 °C.

No.	C (%, *w*/*w*)	C_enzyme_ (µM)	Size (nm)		Z-Average (nm)	PDI	ξ (mV)	EE, %	LC, %
			Intensity	Number					
1	0.1	2.5	106 ± 12	44 ± 8	95 ± 1	0.18 ± 0.01	−5 ± 0.7	89 ± 7	16 ± 1.5
2		5	106 ± 12	51 ± 11	113 ± 1	0.23 ± 0.01	−7 ± 0.6	88 ± 4	32 ± 1
3	0.2	2.5	122 ± 12	44 ± 9	117 ± 4	0.26 ± 0.01	−5 ± 0.7	90 ± 2	8.2 ± 0.2
4		5	106 ± 13	44 ± 9	96 ± 1	0.17 ± 0.01	−6.2 ± 0.1	90 ± 3	16 ± 0.5
5		12.5	122 ± 14	51 ± 11	114 ± 1	0.16 ± 0.02	−5.5 ± 0.2	92 ± 2	41 ± 1
6	0.5	2.5	122 ± 15	44 ± 9	108 ± 1	0.17 ± 0.01	−4.5 ± 0.3	89 ± 2	3.2 ± 0.1
7		5	106 ± 12	44 ± 10	100 ± 1	0.2 ± 0.01	−9 ± 0.5	89 ± 2	6.5 ± 0.1
8		12.5	106 ± 14	59 ± 12	106 ± 1	0.17 ± 0.01	−8.3 ± 0.3	82 ± 7	15 ± 2

**Table 4 ijms-24-15756-t004:** Pharmacokinetic parameters observed in mice after intravenous injection of free enzyme and enzyme-loaded nanoreactors first time and second time 1 month after 1st injections. The dose of enzyme was 3.7 mg/kg. Results represent the mean ± SE for five mice.

Sample	Number of Injections	α (min^−1^)	t_1/2α_ ^1^ (min)
Free enzyme	1, 1 day	0.0315 ± 0.0045	22.00 ± 3.15
Enzyme-loaded nanoreactors	1, 1 day	0.0265 ± 0.0028	26.16 ± 2.83
Enzyme-loaded nanoreactors	2, 30 days	0.0277 ± 0.0025	25.02 ± 2.24

^1^ t_1/2α_ = ln2/α, in which α is the elimination rate (min^−1^) from blood. Data were analyzed via one-way ANOVA (*p* ≤ 0.05).

## Data Availability

Not applicable.

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
