# Peer review of "Tuning the Envelope Structure of Enzyme Nanoreactors for In Vivo Detoxification of Organophosphates"

_ijms, 2023, doi:10.3390/ijms242115756_

Round 1

Reviewer 1 Report

Comments and Suggestions for Authors

This manuscript studied the envelope structure of enzyme nanoreactors for in vivo detoxification of organophosphates. Nanoreactors based on the PEG-PPS block-copolymers with different morphologies were prepared. In vivo experiments showed that the detoxification efficacy of the prepared nanoreactors against paraoxon was improved. The mice treated with this method showed no obvious sequelae after 30 days. However, the following issues should be addressed before the paper is considered suitable for the publication in Int. J. Mol. Sci.

1.      There are some format problems in the full text, please check and correct them. For example, on the page 2, the writing of “–” in “[18,19] and–a” is redundant, please delete it. Most of the text lacks a space between the °C and the preceding number, please add a space.

2.      On page 5, the authors mentioned that “No difference between the control and 1a, 1b nanoreactors was observed for the release of 60% of pNp while a slow release is observed for the residual 20%.” However, it is not obvious in Figure 3. Please give a more accurate explanation of the “slow release”, such as the specific time.

3.      On page 5, the authors mentioned that “For the next step, nanoreactors based on 1b, representing polymersomes, were chosen to encapsulate the enzyme.” However, the advantages and disadvantages of 1a and 1b were not mentioned. Why was 1b chosen to encapsulate the enzyme? please give an explanation.

4.      In “Synthesis polymers for nanoreactor construction” on page 13, the description of the megabyte of the nuclear magnetic instrument is inconsistent. Some are 400 MHZ and some are 600 MHz. Please check again to confirm whether it is correct.

5.      The conclusion should be more concise, refining the key issues solved in this article, rather than listing the experimental results of this article. The authors are suggested to highlight important findings and include afterthought of this work.

6.      In introduction, the authors mentioned “Conversely, drug delivery systems focus on encapsulating drugs and their release under the influence of external factors.” In order to support this statement, the following recently published important related papers should be cited: Exploration 2021, 1, 21; Chin. Chem. Lett. 2023, 108740; Biomed. Eng. Commun. 2023, 2, 15; VIEW. 2023;20220064.

Comments on the Quality of English Language

Moderate editing of English language required

Author Response

Comments and Suggestions for Authors

This manuscript studied the envelope structure of enzyme nanoreactors for in vivo detoxification of organophosphates. Nanoreactors based on the PEG-PPS block-copolymers with different morphologies were prepared. In vivo experiments showed that the detoxification efficacy of the prepared nanoreactors against paraoxon was improved. The mice treated with this method showed no obvious sequelae after 30 days. However, the following issues should be addressed before the paper is considered suitable for the publication in Int. J. Mol. Sci.

  1. There are some format problems in the full text, please check and correct them. For example, on the page 2, the writing of “–” in “[18,19] and–a” is redundant, please delete it. Most of the text lacks a space between the °C and the preceding number, please add a space.

Authors: Text was corrected.

  1. On page 5, the authors mentioned that “No difference between the control and 1a, 1b nanoreactors was observed for the release of 60% of pNp while a slow release is observed for the residual 20%.” However, it is not obvious in Figure 3. Please give a more accurate explanation of the “slow release”, such as the specific time.

Authors: The release of 60% of pNp is very fast and burst release is observed for the control and 1a, 1b nanoreactors. Then the release for next 40% pNp is slow. Full release of pNp (100%) from nanoreactor 1b takes more than 24 hours.

  1. On page 5, the authors mentioned that “For the next step, nanoreactors based on 1b, representing polymersomes, were chosen to encapsulate the enzyme.” However, the advantages and disadvantages of 1a and 1bwere not mentioned. Why was 1bchosen to encapsulate the enzyme? please give an explanation.

Authors: Thus, nanoreactor 1b with vesicular morphology and encapsulation efficiency and loading capacity for hydrophilic compound pNp was selected for encapsulation of the enzyme.

  1. In “Synthesis polymers for nanoreactor construction” on page 13, the description of the megabyte of the nuclear magnetic instrument is inconsistent. Some are 400 MHZ and some are 600 MHz. Please check again to confirm whether it is correct.

Authors: NMR spectra were recorded on 400 MHz [400.1 MHz (1Н), 100.6 MHz (13С)] and/or on 600 MHz [600.1 MHz (1Н), 150.9 MHz (13С)] spectrometers. It is a correct information.

  1. The conclusion should be more concise, refining the key issues solved in this article, rather than listing the experimental results of this article. The authors are suggested to highlight important findings and include afterthought of this work.

Authors: We expanded the conclusion and highlighted the interest of enzyme nanoreactor therapy against other types of OPs.

  1. In introduction, the authors mentioned “Conversely, drug delivery systems focus on encapsulating drugs and their release under the influence of external factors.” In order to support this statement, the following recently published important related papers should be cited:Exploration 2021, 1, 21; Chin. Chem. Lett.2023, 108740; Biomed. Eng. Commun20232, 15; VIEW. 2023;20220064.

Authors: They incorporated, now ref.16-19.

Reviewer 2 Report

Comments and Suggestions for Authors

The manuscript is written well and the experiments are organized in a good way. The authors claimed that nanoreactors based on the PEG-PPS block-copolymers with different morphologies for the studies in 2 vivo detoxification of organophosphates. The manuscript can be published with some minor revisions. 

The minor comments are detailed below,

1. It is better to add a high-resolution image of TEM in supporting information.

2. The FTIR spectra of copolymer 1a and 1b shown in Figures S4 and S7 can be merged together to show the exact difference in functional groups.

Author Response

The manuscript is written well and the experiments are organized in a good way. The authors claimed that nanoreactors based on the PEG-PPS block-copolymers with different morphologies for the studies in 2 vivo detoxification of organophosphates. The manuscript can be published with some minor revisions. 

The minor comments are detailed below,

  1. It is better to add a high-resolution image of TEM in supporting information.

Authors: Images of TEM were added in supporting information file.

  1. The FTIR spectra of copolymer 1a and 1b shown in Figures S4 and S7 can be merged together to show the exact difference in functional groups.

Authors: The FTIR spectra of copolymer 1a and 1b shown in Figures S4 and S7 do not contain different functional groups, whose absorption bands could be compared. The difference between the polymers is in the number of ethylene glycol and propylene sulfide moieties. Nevertheless, we have merged these spectra together.

Reviewer 3 Report

Comments and Suggestions for Authors

This is an extraordinarily important paper.  However, it is flawed by its presentation. I suggest that the authors have a colleague who is not familiar with their work read it to check for presentation problems. 

Here are the issues that I have found. Text in italics is copied from the paper. 

Medical applications of enzymatic nanoreactors and nanodevices [1–5] are exponentially growing up. line 35

The use of the word exponential should be proven by data. It is best not to use that word, since it is so often misused. Also, growing up is poor English usage. 

The creation of nanoreactors for trapping toxicants involves the complete sealing of enzyme molecules inside and long-term circulation in the body to trap and neutralize toxicants present in the bloodstream and slowly released from depot sites [2,16,17]. Lines 47-49

I thought that the toxicants in this paper were being destroyed -that is, neutralized -  by the enzyme present in the nanoreactors. Are they also trapped in the nanoreactors. This sentence confuses the issue. 

Attempts to use encapsulated organophosphate (OP)-reacting enzymes for stoichiometric or catalytic inactivation of OPs have been undertaken for more than 20 years and led to very effective formulations [18,19] and a novel nanoscavenger devices for OP detoxification [20,21].  Lines 50-53

I see no evidence in the citation list suggesting that use of encapsulated organophosphate (OP)-reacting enzymes have been used for over 20 years. Please cite this number or correct it. 

In a previous work [27] we showed that injection of enzymatic nanoreactors (E-nRs) containing a high concentration of an evolved quintuple mutant of an archea PTE from Saccharolobus solfataricus [28] can protect mice against multiple LD50 of paraoxon (POX) Lines 61-63

There is no explanation, at this point in the paper or prior to this point, what the abbreviation PTE stands for. Archae should be archaeal. This problem is present at several points in this paper. 

PEG guarantees stealth behavior for improved circulatory properties after systemic administration. Lines 69-70

What does this sentence mean? I can only guess. Also, I assume that PEG = polyethylene glycol? 

Table I is missing any explanation of what it purports to display. 

C?      Z-average?           PDI?           ξ  ??

The reader should not be forced to look elsewhere in the paper for this information. 

The POX concentration in blood can reach 6 µM at real field conditions of severe poisoning. Line 186

This sentence needs a citation.

It is important to work at enzyme concentrations exceeding the POX concentration both for the possibility of self-diffusion of POX into nanodevices and for the reaction of POX with the enzyme inside the reactor under second-order conditions. Lines 187-188

The word concentrations in this sentence is ambiguous. Does it mean molar concentrations? If so, why do the enzyme concentrations need to exceed those of POX? Is there no turnover of the enzyme in this situation? Is POX an irreversible inhibitor of the PTE from Saccharolobus solfataricus? If so, that should be stated explicitly. 

t1/2α = ln2/α, in which α is the distribution rate (min-1 ) from blood. Data were analyzed via one-way Line 298

What do the words distribution rate mean in this context?

 p-nitrophenol (pNp) 99% pure was from Alfa Aeser, Karlsruhe, German

Line 405

There are two spelling mistakes in this sentence. Correct are Alfa Aesar  and Germany. Also, I understand that the corporate name of Alfa Aesar has changed or is changing. Not that is is important in this context. 

In the equations given on lines 510-511. What are the units of the word "amount"?

This paper deals with the insecticide paraoxon. One of the distributors of paraoxon describes it as "a surrogate of chemical warfare agents."  (See https://www.scbt.com/p/paraoxon-311-45-5)

Is there any literature to suggest that the PTE from Saccharolobus solfataricus also effective against Sarin, the agent reported to have been used a decade ago? See https://www.state.gov/tenth-anniversary-of-the-ghouta-syria-chemical-weapons-attack/

Thus, do the methods described in this paper have the potential to be protective against military nerve warfare agents?  This topic should be addressed, at least briefly. 

Comments on the Quality of English Language

Please see the comments in the section above. In general, the English language usage is very good. There are a few isolated problems that are listed above. 

Author Response

Comments and Suggestions for Authors

This is an extraordinarily important paper.  However, it is flawed by its presentation. I suggest that the authors have a colleague who is not familiar with their work read it to check for presentation problems.

Here are the issues that I have found. Text in italics is copied from the paper. 

Medical applications of enzymatic nanoreactors and nanodevices [1–5] are exponentially growing up. line 35

The use of the word exponential should be proven by data. It is best not to use that word, since it is so often misused. Also, growing up is poor English usage.

Authors: It was corrected. Medical applications of enzymatic nanoreactors and nanodevices [1–5] are rapidly developing.

The creation of nanoreactors for trapping toxicants involves the complete sealing of enzyme molecules inside and long-term circulation in the body to trap and neutralize toxicants present in the bloodstream and slowly released from depot sites [2,16,17]. Lines 47-49

I thought that the toxicants in this paper were being destroyed -that is, neutralized -  by the enzyme present in the nanoreactors. Are they also trapped in the nanoreactors. This sentence confuses the issue. 

Authors: The sentence was corrected for clarification. The creation of nanoreactors for trapping toxicants involves the complete sealing of enzyme molecules inside nR body and must insure long-term circulation in the body to trap and neutralize toxicants present in the bloodstream and may be slowly released from cellular and organ depot sites [2,16,17].

The last part of this sentence refers to the fate of toxic molecules in the body: after penetration in the body, toxic molecules circulate in the blood stream and may also accumulate in depot sites, e.g., OP molecules may accumulate in fat from where they are subsequently slowly released in the blood stream again.

Attempts to use encapsulated organophosphate (OP)-reacting enzymes for stoichiometric or catalytic inactivation of OPs have been undertaken for more than 20 years and led to very effective formulations [18,19] and a novel nanoscavenger devices for OP detoxification [20,21].  Lines 50-53

I see no evidence in the citation list suggesting that use of encapsulated organophosphate (OP)-reacting enzymes have been used for over 20 years. Please cite this number or correct it. 

Authors: It was corrected. Attempts to use organophosphate (OP)-reacting enzymes and their encapsulated forms for stoichiometric or catalytic inactivation of OPs have been undertaken for more than 20 years and led to very effective formulations [18,19] and a novel nanoscavenger devices for OP detoxification [20,21].

In a previous work [27] we showed that injection of enzymatic nanoreactors (E-nRs) containing a high concentration of an evolved quintuple mutant of an archea PTE from Saccharolobus solfataricus [28] can protect mice against multiple LD50 of paraoxon (POX) Lines 61-63

There is no explanation, at this point in the paper or prior to this point, what the abbreviation PTE stands for. Archae should be archaeal. This problem is present at several points in this paper. 

Authors: In a previous work [27] we showed that injection of enzymatic nanoreactors (E-nRs) containing a high concentration of an evolved a multiple mutant of the hyperthermophilic archaea Saccharolobus solfataricus phosphotriesterase (PTE)-like lactonase (PLL) [28] can protect mice against multiple LD50 of paraoxon (POX) (up to 16.6 LD50 as the sole prophylaxic mean).

PEG guarantees stealth behavior for improved circulatory properties after systemic administration. Lines 69-70

What does this sentence mean? I can only guess. Also, I assume that PEG = polyethylene glycol? 

Authors: It was corrected. Polyethylene glycol (PEG) guarantees stealth behavior for improved circulatory properties of nanoparticles after their systemic administration.

Table I is missing any explanation of what it purports to display. 

C?      Z-average?           PDI?           ξ  ??

The reader should not be forced to look elsewhere in the paper for this information.

Authors: It was corrected. Where size is hydrodynamic diameter, Z-average is the mean size, PDI is polydispersity index, ξ or zeta potential is electrokinetic potential.

The POX concentration in blood can reach 6 µM at real field conditions of severe poisoning. Line 186

This sentence needs a citation.

Authors: The reference was added Eyer, F.; Eyer, P. Enzyme-based assay for quantification of paraoxon in blood of parathion poisoned patients. Hum. Exp. Toxicol.

1998, 17, 645–651.

It is important to work at enzyme concentrations exceeding the POX concentration both for the possibility of self-diffusion of POX into nanodevices and for the reaction of POX with the enzyme inside the reactor under second-order conditions. Lines 187-188

The word concentrations in this sentence is ambiguous. Does it mean molar concentrations? If so, why do the enzyme concentrations need to exceed those of POX? Is there no turnover of the enzyme in this situation? Is POX an irreversible inhibitor of the PTE from Saccharolobus solfataricus? If so, that should be stated explicitly. 

Authors:  The word concentration refers to molar concentration of encapsulated enzyme. The enzyme concentration is higher than the highest expected toxicant concentration so that the enzyme nanoreactor works under second order conditions. Under these conditions, there is enzyme turnover as for enzyme kinetics at high enzyme concentration (this point is developped in ref 26). POX is not irreversible inhibitor of POX, it is very good substrate of the enzyme.

t1/2α = ln2/α, in which α is the distribution rate (min-1 ) from blood. Data were analyzed via one-way Line 298

What do the words distribution rate mean in this context?

Authors: Distribution rate is elimination rate from blood stream. We corrected to “alpha is the elimination rate (min-1) from blood.”

 p-nitrophenol (pNp) 99% pure was from Alfa Aeser, Karlsruhe, German

Line 405

There are two spelling mistakes in this sentence. Correct are Alfa Aesar  and Germany. Also, I understand that the corporate name of Alfa Aesar has changed or is changing. Not that is is important in this context. 

Authors: It was corrected. p-nitrophenol (pNp) 99% pure was from Alfa Aesar, Karlsruhe, Germany

In the equations given on lines 510-511. What are the units of the word "amount"?

This paper deals with the insecticide paraoxon. One of the distributors of paraoxon describes it as "a surrogate of chemical warfare agents."  (See https://www.scbt.com/p/paraoxon-311-45-5)

Is there any literature to suggest that the PTE from Saccharolobus solfataricus also effective against Sarin, the agent reported to have been used a decade ago? See https://www.state.gov/tenth-anniversary-of-the-ghouta-syria-chemical-weapons-attack/

Thus, do the methods described in this paper have the potential to be protective against military nerve warfare agents?  This topic should be addressed, at least briefly. 

Thus, do the methods described in this paper have the potential to be protective against military nerve warfare agents?  This topic should be addressed, at least briefly.

Authors: This SsoPox variant was shown to efficiently degrade chemical warfare nerve agents analogues (Jacquet et al. 2023) but was not tested against real agents, however other previously reported SsoPox variants were shown active against tabun, sarin, soman and cyclosarin (Merone et al. 2010; Suzumoto et al. 2020).

Jacquet, Pauline, Raphaël Billot, Amir Shimon, Nathan Hoekstra, Céline Bergonzi, Anthony Jenks, Eric Chabrière, David Daudé, et Mikael H. Elias. 2023. « Changes in Active Site Loop Conformation Relate to the Transition toward a Novel Enzymatic Activity ». bioRxiv. https://doi.org/10.1101/2023.05.22.541809.

Merone, Luigia, Luigi Mandrich, Elena Porzio, Mosé Rossi, Susanne Müller, Georg Reiter, Franz Worek, et Giuseppe Manco. 2010. « Improving the Promiscuous Nerve Agent Hydrolase Activity of a Thermostable Archaeal Lactonase ». Bioresource Technology 101 (23): 9204‑12. https://doi.org/10.1016/j.biortech.2010.06.102.

Suzumoto, Yoko, Orly Dym, Giovanni N. Roviello, Franz Worek, Joel L. Sussman, et Giuseppe Manco. 2020. « Structural and Functional Characterization of New SsoPox Variant Points to the Dimer Interface as a Driver for the Increase in Promiscuous Paraoxonase Activity ». International Journal of Molecular Sciences 21 (5): 1683. https://doi.org/10.3390/ijms21051683.

Round 2

Reviewer 3 Report

Comments and Suggestions for Authors

Table 2S in the supplement needs further explanation. How much paraoxon is associated with the controls?

Comments on the Quality of English Language

In Line 35 the sentence Medical applications of enzymatic nanoreactors and nanodevices [1–5] are rapidly developing.
is poorly written. Perhaps
are rapidly being developed
is better usage.  

in Line 67, This SsoPox variant was shown to efficiently degrade chemical warfare nerve agents analogues 

is better stated as 
This SsoPox variant was shown to efficiently degrade analogs of chemical warfare nerve agents.

Author Response

Comments and Suggestions for Authors

Table 2S in the supplement needs further explanation. How much paraoxon is associated with the controls?

Authors: Caption of Table S2 and section 3.7.2. were clarified. Prophylaxis and post-exposure treatment of paraoxon s.c. acute toxicity by i.v. administration of enzyme-loaded nanoreactors in mice, where Control 1 is the EtOH 10% in sodium chloride 0.9% solution (s.c.) and Control 2 is empty nanoreactor solution (i.v.).

Comments on the Quality of English Language

In Line 35 the sentence Medical applications of enzymatic nanoreactors and nanodevices [1–5] are rapidly developing. is poorly written. Perhaps are rapidly being developed is better usage.  

Authors: It was corrected. Medical applications of enzymatic nanoreactors and nanodevices [1–5] are rap-idly being developed

in Line 67, This SsoPox variant was shown to efficiently degrade chemical warfare nerve agents analogues is better stated as 
This SsoPox variant was shown to efficiently degrade analogs of chemical warfare nerve agents.

Authors: It was corrected. This SsoPox variant was shown to efficiently degrade analogs of chemical warfare nerve agents.
